# Viaduct and Bridge Structural Analysis and Inspection through an App for Immersive Remote Learning

Antonino Fotia [1] and Vincenzo Barrile [2,*]

1   PAU Department of Heritage Architecture and Urbanism, Mediterranean University,
    89124 Reggio Calabria, Italy
2   DICEAM Department of Civil, Energy, Environmental and Material Engineering, Mediterranean University,
    89124 Reggio Calabria, Italy
*   Correspondence: vincenzo.barrile@unirc.it

**Abstract:** Until now, in the design phase of infrastructures there has been a general tendency to "economize" the resources allocated to them. This modus operandi did not consider the installation of monitoring and control systems as an integral part of the infrastructure itself, not considering the high post-intervention costs. This work aims to show how the integration of immersive technologies, including Virtual/Augmented/Mixed Reality, combined with geomatics, survey and structural monitoring techniques can ensure a better visualization and understanding of the different contexts in which the managing bodies are required to guarantee maintenance interventions. In particular, the potential of an application, developed by the authors in Unity 3D, to help the managing institution is described. The app permits the user to explore infrastructures under inspection in a virtual environment. This makes all the information related to the infrastructure available and accessible through the 3D analysis (which is manageable in the app after a mesh edge reduction phase) exploiting the full potential of Mixed/Virtual Reality. The main ability of our application derives from the chance to easily use and integrate different techniques (3D models, information models for construction, VR/AR) allowing for the choice of different 3D models testing and performing their simplification and dimensional reduction. This makes the loading phase of the application faster and the user experience easier and better. The experimentation of the proposed methodology was conducted on a viaduct located in Reggio Calabria.

**Keywords:** augmented reality; virtual reality; 3D model; remote learning; structural analysis

## 1. Introduction

The Italian infrastructure heritage (bridges, viaducts) is relatively old. It is not uncommon for project documentation (relating to infrastructures built more than 50 years ago) to not be available or to be lost. The managing institution in charge of road infrastructure maintenance cannot investigate by comparing with the project data [1], but they must conduct surveys and inspections.

Until now, in the design phase of the infrastructures, there was a general tendency to "economize" the resources allocated to them. This modus operandi did not consider the installation of monitoring and control systems as an integral part of the infrastructure itself, not considering the high post-intervention costs.

To date, the use of sensors and traditional inspection systems is not very convenient from an economic point of view or taking into account the cost of installation [2,3]. Moreover, the results of the traditional methodologies used today are not adequately grouped or available on integrated platforms. Clearly, there are web platforms and GIS systems, but their use is aimed at representing results related to a single problem.

The proposed methodology, therefore, concerns the design and subsequent implementation of a system to monitor infrastructures, which integrates soft computing and geomatic

methodologies with the purpose of introducing automation in the collection, processing, forecasting and transmission of data and especially the possibility of remote inspection and virtual consultation in situ through the techniques of Mixed, Augmented and Virtual Immersive Reality. This is exactly the knowledge gap to be covered by the research, through information digitization and the consequent possibility to have them available at any time, as well as through innovative methodologies such as Augmented/Virtual Reality (used, for example, to visualize some parts of interest in 3D). From a methodological point of view, the results were obtained through the integration of different applications within the same environment for the development of an app (topographic surveys, 3D modeling data collection and analysis, etc.) for mobile devices, laptops and HoloLens.

If we consider the level of development achieved by the software to date, from an app implementation point of view, it is quite clear that the main difficulty in creating an app in practical terms lies precisely in its conception, in the choice of data to be used and in the refinement of the results (in addition, of course, to the optimization of the spaces necessary for data storage). The developed app has mainly been designed to perform the function of infrastructure monitoring, understood as the use of data relating to singular points of the road network. Thus, it plays the role of a container of 3D models, level of deterioration images, results of structural analysis, maintenance interventions, analysis on the viability, and so on. Among the potential of the developed app, the integration and visualization of these results also through Augmented Reality (AR), Virtual Reality (VR) and Mixed Reality (MxR) are particular interesting, allowing us to directly access data through the framing of the scene by the camera of the device used.

The acquired data (necessary for the server's population to which the app refers) come from individual interdisciplinary analyses and methodologies combined with each other (in particular, geomatics and structural analysis); however, although their management clearly require a purely IT contribution, this process is now manageable even by personnel not really in the IT sector through graphical interfaces or interactive user-device processes.

That said, in this specific case, the realization of this methodology was based on the combination of several models:

- A UAV 3D model detected by drone (initial state of the infrastructure).
- A structural model (scenario n).
- A final model integrated with data acquired over time from various sources and in real-time, directly from any sensors installed on site.

As far as the possibility of remote inspection is concerned, it should be emphasized that in recent years, we are witnessing an increasing diffusion of the BIM (Building Information Modeling) methodology [4], used by the author to represent all phases of the structure's life using a query that applies to the model itself [5], taking into account that the change made to a single "layer" is automatically propagated to the entire model [6].

A process originally conceived for new constructions can be a valuable tool for existing buildings with the right adaptation.

Moreover, the role of Virtual Reality can be applied to 3D modeling, permitting better interaction among the various stakeholders during the process [7]. The use of the model can be improved and simplified using VR [8], for example, allowing the operator to monitor all the operational phases of the work (design, construction, maintenance) [9].

From an application point of view, the AR and VR methodologies are similar and integrable, but different and usable in different application fields. As can be seen from the literature [10], in the field of construction engineering, to which the authors' work refers, in the case of AR, the technology is best suited for displaying renovations and retrofit works, as it combines the real environment with virtual objects. Mutis and Ambekar [11] studied existing AR challenges to implement the project's detailed procedures and presented a system capable of visualizing future interventions on construction sites. AR has also been used to support virtual tours and in situ walkthroughs. Kim et al. [12] presented an AR system to enrich cultural heritage tours by overlaying virtual objects on cultural sites. Tan

and Lim [13] presented an AR virtual tour system to enhance the experience of exploring historical sites. Andri et al. [14,15] presented a system to support campus tours using AR.

With regard to VR, it has been extensively tested for stakeholder engagement in the real estate sector, e.g., [16,17], as it is an ideal means for immersing stakeholders in a virtual environment, helping them to understand what the final product will look like. Kini and Sunil [18] presented a VR system to visualize different types of sustainable building techniques in rural communities, and Xia [19] presented the development of a VR system for virtual campus walkthroughs. Most recent research has focused on providing multi-user functionality. For example, Du et al. [20] presented a multi-user VR platform that allowed different parties (e.g., clients, architects, engineers and general contractors) to interact in a unified VR environment improving stakeholder engagement and communication, and Lin et al. [21] presented a VR approach to improve communication between design teams and healthcare stakeholders.

In addition, Lin et al. [21] highlight that Virtual Reality (VR) technology can be applied as a complement to three-dimensional modeling, leading to better communication in both professional training and professional practice [22]. VR technology can provide virtual reality simulation functions and an interactive experience environment, improving the ability to examine and control the design model; on the other hand, VR can interactively explore in real time and improve cognition, communication and integration in the initial design phase [23]. Recently, most research has adopted VR in plant or building designs, which have focused on visualization, analysis and evaluation at the design and construction stages [24]. Recently, BIM integrated with VR has been one of the most promising recent developments in AEC construction projects [25].

The model can also show the geometric details and technical information that must accompany each construction step. Thus, operators of managing institutions can better understand the information acquired from the available documentation (i.e., the possibility to obtain 3D models corresponding to different states of shape, simulating distinct phases, or the possibility to directly view any connections of elements that make up the infrastructure—structural elements, services, etc.) and can order them temporally (4D models) [26,27].

In fact, Virtual Reality (VR) completely immerses learners (users) in alternative digital worlds. Content is accessed through VR headsets such as an HTC Vive or Oculus Quest, often combined with headphones and hand controllers that allow the learner to navigate their way around their virtual space. Rather than blocking out the real world, Augmented Reality blends it with digital content. Digital assets can take many shapes and forms, so it can be flat and 2D, which is great for instructional information, or be more complex and 'real' in 3D. Content can be triggered by specific objects or geographical places. Mobile devices, such as smartphones and tablets, allow the learner to access content, making it easily accessible. Mixed Reality combines elements of Virtual and Augmented Reality. Like Augmented Reality, it overlays digital content with the real world. This content is anchored to (and interacts with) objects in the real world. A major difference between Mixed and Augmented Reality is that in Mixed Reality, digital assets can be visibly obscured by real-world objects [28]. Immersive learning is a hugely effective way for many users to develop their knowledge and skills. It provides artificial, digitally created content and environments that accurately replicate real-life scenarios so that new skills and techniques can be learned and perfected. Users are not simply passive spectators; they get to be active participants who directly influence outcomes [29].

The experimentation of the proposed methodology guarantees that infrastructure managers can transfer know-how to the various technicians and operators in a simplified or, when necessary, more in-depth way.

Italian legislation has established guidelines that constitute an important methodological basis for the management bodies and technicians responsible for assessing the vulnerabilities and critical issues related to the structural safety of existing bridges, to standardize a common and shared approach of analysis, monitoring and choice of intervention times. The purpose of visual inspection is to map the bridge's degradation and its state

of conservation. The points of degradation and defects of the structure must be recorded in the appropriate defect cards [30]. Visual inspections are not very objective and are not standardized, as it is an operator who fills the cards. The proposed methodology makes it easy to inspect and verify by third parties what was detected and reported in the data sheets by the technicians, thanks also to the potential of Mixed and Augmented Virtual Reality. The experimentation of the proposed methodology was carried out on a motorway viaduct located in the city of Reggio Calabria (Figure 1).

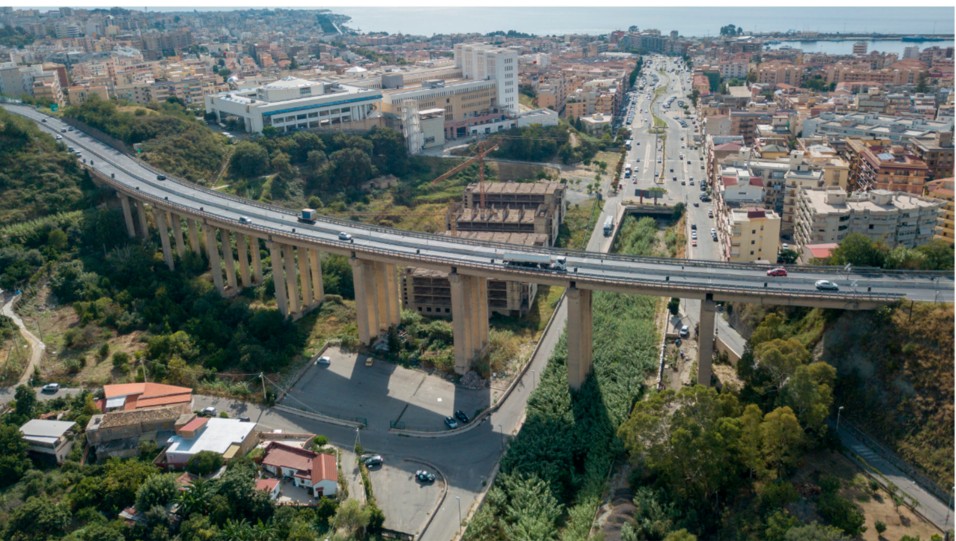

**Figure 1.** Annunziata Viaduct, Reggio Calabria, Italy.

The viaduct was built in pre-stressed reinforced concrete, over the river "Annunziata" in 1970. The entire structure consists of two bridges (one in each direction) supported by a pair of piles with a common base. The viaduct (25 m high and 254 m long) has nine short spans (27 m long) with four beams and three pre-stressed reinforced concrete crosses, and it is curved with a curvature radius of 150 m. The piles have a rectangular section of 2.50 m × 1.60 m, and the dimensions of the column bases are 8 m × 3 m [31].

The infrastructure (managed by ANAS S.p.A—"Azienda Nazionale Autonoma delle Strade", an Italian company that deals with road infrastructure and manages the network of state roads and highways of national interest) is part of the A2 motorway and is exposed to high seismic risk. The position of the infrastructure is strategic and fundamental; in fact, it connects the north and south of the city, allowing the circulation of vehicles and trucks outside the city. In the event of a collapse, the entire motorway would be interrupted with elevated risk and consequences on the circulation of vehicles and on the response to emergencies.

In this context, the app stands as the main e-learning tool [32]. Immersive learning is a technique that creates artificial content and environments that accurately replicate real-life scenarios. In this environment, users are no longer simply passive spectators, but become active participants who interact with the surrounding environment by simulating behaviors, decisions and mistakes even before applying them in real working life.

Immersive learning, therefore, offers a safe and risk-free space where learning can be repeated, and success can be accurately measured, but not only that: immersing employees in "real" work situations improves both their involvement and the long-term retention of the acquired knowledge [33].

## 2. Materials and Methods

To date, the level of development reached by software for creating apps is quite advanced; moreover, the implementation of any code is accessible to all, through functions and buttons. In fact, app creation software is generally made available by third-party companies and allows users to develop content within an integrated environment. It is easier to use

than Software Development Kit SDKs and allows users to work on a single project that can be exported to various formats, thus ensuring compatibility with various platforms. So, from an app implementation point of view, the main difficulty in creating an AR/VR app, in practical terms, lies precisely in its conception, in the choice of data to be used and in the refinement of the results (in addition of course to the optimization of the spaces necessary for data storage). The design of an app should be made user-friendly, especially if it includes learning methods so as to incorporate digital game-based learning. Learning game apps can be divided into two categories: games and simulators. In a game, we have goals and levels, and our advancement depends on successful completion. This provides an indirect measure for the skills and knowledge that the user is acquiring. In a simulator, one learns by interacting with a computerized environment designed to act out the very environment in which one will eventually apply one's skills and knowledge. The processes and methodologies used for data acquisition and for the creation of the app are, therefore, reported below.

### 2.1. Survey and Structural Model Implementation

One of the main aspects related to structural monitoring is the prediction of collapse. The structural engineer uses the data from the inspections to develop a model, to predict the temporal development of the damage (if any) and the eventual collapse.

As it is known, to have information on the structural behavior and on the collapse behavior of bridges and viaducts, one-dimensional finite element models (FEM) can be used; if more detailed information is required, more complex analyses must be conducted, but this causes an increase in processing times. The response of the structural model varies according to the load combinations and the degree of degradation assumed for the infrastructure, generating different responses to the changing scenarios predicted (state of interest service limit and last state limit). In the case of existing works, structural monitoring integrates the data utilized in the design phase. It is also possible to determine the response of the structure combined with data obtained from surveys in situ and data from tests carried out in the laboratory. In this case, therefore, we are aware of the response and loading actions, but what we want to determine is the pattern. It could be said that in the first case, we deal with a direct problem, while in the second case, we deal with the opposite problem. The creation of a final model can be used to analyze the dynamic structural response and the structure's behavior changing the applied loads of the boundary conditions, and the characteristic parameters of materials [34]. The model can be obtained by combining the monitoring information and the results of geomatic surveys (project drawings, UAV surveys, laser scanner surveys, etc.) with tests on materials and structural monitoring. To validate the final obtained model, the stresses detected are then compared with those found during the in situ tests. For example, the actions acting on the model from the project were defined through the TGM (average daily traffic from the normative), while the actual number of vehicles (through pattern recognition system in the various frames) are actually counted in situ (during peak hours, therefore, of maximum load), or the reduction of the section is measured through photogrammetric techniques and validated through in situ measurements. Clearly, this can lead to a change in structural behavior (degradation). Therefore, regarding the response of the structural model, the various parameters are used to update the models already built and to obtain answers that are more congruous with the situation (an activity that nowadays is still carried out in a non-automated way, through the use of operators who visually inspect bridges and viaducts with destructive investigations). The comparison between the model (at the present time and the simulated one at the time n) allows the users to identify any anomalies (subsidence, increase in degradation, changes, etc.); therefore, it is possible to decide a priori how to intervene to prevent the occurrence of any damage, carrying out preventive maintenance activities that clearly reduces and spreads the funds available on the entire road infrastructure. Four different levels of degradation were considered and inserted by CSV file into the software. In detail,

degradation was considered by the introduction of indexes of cracking and corrosion. If uniform corrosion is considered, the damage indexes listed below can be taken into account.

### 2.2. Augmented, Virtual and Mixed Reality in Unity Environment

Virtual Reality is a tool that has been strongly developing in recent years and that would, on the one hand, allow us to visualize in a virtual environment the results obtained from the various analyses in the different scenarios and, on the other hand, to manage all the information available (cataloguing them and making them available to users with particular reference to the managing authorities). While VR and AR share many similar technologies, such as various tracking sensors and displays, they represent two different approaches in blending the physical and virtual world realities. VR and AR are defined as follows:

- Virtual Reality (VR): "an artificial environment which is experienced through sensory stimuli (such as sights and sounds) provided by a computer and in which one's actions partially determine what happens in the environment" [35].
- Augmented Reality (AR): "AR allows the user to see the real world, with virtual objects superimposed upon or composited with the real world. Therefore, AR supplements reality, rather than completely replacing it" [36].

AR traditionally overlays digital content onto a live view of the environment; moreover, Virtual Reality (VR) and Augmented Reality (AR) are among the technologies that will play a central role in building new business models, breaking down barriers in communication and modernizing professions in every sector, as well as designing user experiences. The former (VR) enables multisensory experiences aimed at exploring 3D digital environments and interacting with the elements that compose them. The latter (AR) enriches the real world with digital visual overlays of text, images and 3D models, useful for example in providing the user with extra information about a product on display, a location, a commercial activity or a procedure to follow [37]. When the user can see associated Augmented Reality information superimposed in Virtual Reality scenarios, we speak instead of Mixed Reality [38,39]. Therefore, to summarize, AR/VR/MR methodologies can mix and integrate with each other even if they are different. Figure 2 shows a schematization of the three methodologies (used by Microsoft during a presentation of the HoloLens) that describes and identifies them in accordance with the ideas of the authors.

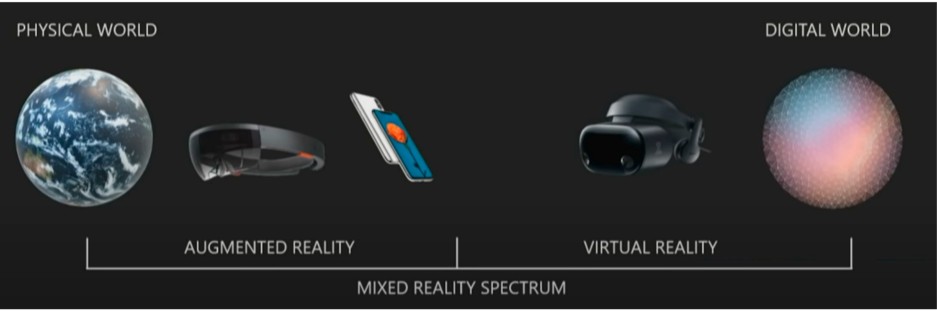

**Figure 2.** The schematization of interaction in the virtual world–real world.

From this, it can be deduced that Mixed Reality straddles Augmented Reality and Virtual Reality, mixing the two methodologies and, therefore, allowing information to be received in a different way. They consider Mixed Reality as the spectrum between digital and physical which makes it a helpful kind of category; in fact, this does not just encompass Virtual Reality and augmented reality, it also encompasses everything in between.

Regarding the implementation of Augmented/Virtual and Mixed Reality applications, Unity 3D is the most used platform. It uses the Unity Script programming language (developed by integrating two types of programming languages: JavaScript and C#). Scripts, known in Unity as behaviors, allow the user to capture resources in scenes and make them interactive. Multiple scripts can be linked to a single object, allowing for easy

code reuse. The app realization phase is also facilitated by a series of tools that allow the users to integrate the programming part with the graphic needs. Virtual and Augmented Reality have inspired the research world in the field of Human–Machine Interface (HMI) systems, hardware and/or software components. Thanks to them, users can interact with the virtual world (smartphone display through a camera, graphic interface of an app, microphone to give voice commands, mouse and keyboard connected to the PC and wearable devices, such as bracelets and gloves equipped with sensors, able to recognize human gestures and make tactile experiences possible in virtual worlds). The Microsoft Holo Lens team (to facilitate the connection between the generated app and the headset provided by them—the HoloLens) has made available a collection of custom tools: The Mixed Reality Toolkit that can be downloaded from GitHub. This allows the users to define the position of the user's point of view in the virtual scene and superimpose it on top of the HoloLens camera [40,41]. In particular, HoloLens2 through JoinXR allows multiple users to interact within the same scene. For example, in a digital twin of a building, we can see that in the mixed reality environment, as we move around, the building stays in exactly the same place as if it was physically in the room with us. It can be thought of as a three-dimensional PowerPoint. We can bring the content of various types into the joining platform and arrange it and compose it as flexibly as we would in something like PowerPoint. The key difference is that it is necessary to use, for example, Azure Remote rendering. There is no limit to the complexity of the models that it is possible to display. Essentially, we can display anything in 3D and organize it as effectively as we would a PowerPoint. This is a key point, especially when we are talking about BIM data or a full digital twin of a building because the file is quite large. With Azure Remote rendering, we are able to bring in the actual file, the true digital twin of the building, which is very powerful. We are both physically in the same room, but we do not have to be. We could work in a hybrid setting. The way we represent ourselves in these 3D meetings is by using avatars, an easy way to represent ourselves in a 3D space. The position of the head and hands are tracked so that it is possible to fully express body language as we move around and interact with everyone else in the meeting. The audio is spatial. So, when somebody speaks, it goes into the left ear of someone if you are standing to the left of them, etc.

The experience with visors and wearable devices can still be shared with other users through the projection on the screen of the user's actions in a virtual environment [42]. Once the developed environment has been defined, we proceed with the 3D model importation, associating the scripts and the basic settings to ensure interaction with it through tags and labels. The Mixed Reality Toolkit includes other scripts and tools to enhance the MR experience. The toolkit enables gesture and gaze-based interaction [43].

The specific achievements of the authors concern the creation of an app that integrates the methodologies described above in order to create a tool available to the technicians of the managing authorities that is able to act as a container of information, made available at any time (even during on-site inspections). In particular, the use of Virtual Reality allows the user to be able to view the work in its temporal progression, reporting the interventions already carried out and those planned, in order to highlight any overlaps of any kind. The use of Virtual Reality instead would make it possible for various information to be immediately available on the individual components and/or services or the data of events that interact to some extent with the infrastructure itself (for example, the number of accidents, average daily traffic, number of heavy vehicles, etc.). In particular, from a methodological point of view, an app has been developed on the Unity platform for the management of territorial infrastructures. The app guides the end user (user/administration) to surveys and monitoring infrastructures (as a whole, in its parts and in the context in which it is inserted), offering quality content in a simple and intuitive way. The application is still in the experimentation and verification phase, which integrates 3D geo-topographic models reconstructed through photogrammetric techniques, simplified through methods of reduction of faces and vertices that allow the quality and precision of the model to be maintained. This results in a tool that can be used to extract geometric information. Unity platform

programming is based on "GameObjects" [44,45]. GameObjects are the fundamental objects in Unity that represent characters, props and scenery. They do not accomplish much in themselves, but they act as containers for components which implement the functionality. Every object in the scene is a GameObject, from characters and collectable items to lights, cameras and special effects. However, a GameObject cannot do anything on its own; we need to give it properties before it can become a character, an environment or a special effect. GameObjects can be with or without graphic representation. "Mono-Behavior" scripts are associated with them, which allows for the presence of new data in the buffer to be verified. Thanks to the technology developed in recent years, it is possible to visualize the results of the applications of increased Visual and Mixed Reality, as well as through specially designed devices. In fact, in the developed app, the users can interact with the surrounding environment through the screen of the device or through Microsoft HoloLens [46] (Figure 3).

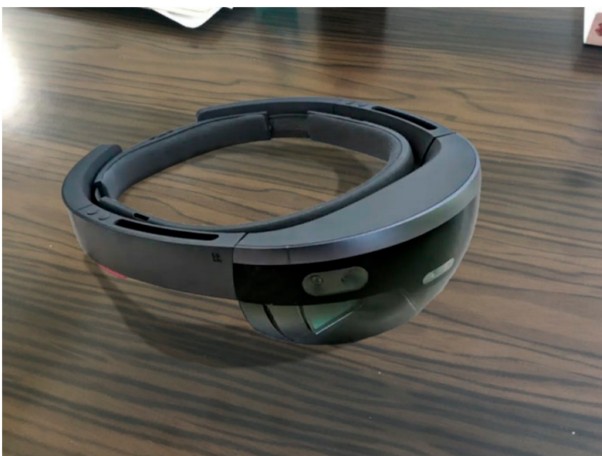

**Figure 3.** Microsoft HoloLens.

The Microsoft HoloLens is a transparent head-mounted optical display (HMD) designed for AR/MR interactions, by looking, gesticulating and using one's voice. For example, the user can concentrate on the object that is seen [47,48] via head-tracking. The implementation of the Microsoft HoloLens has provided the phases and commands shown in the following diagram (Figure 4) [49]:

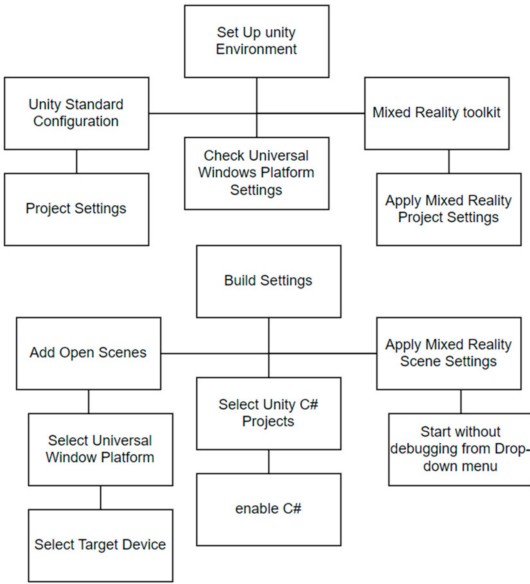

**Figure 4.** Flow Chart Unity Command.

### 2.3. Photogrammetric 3D Model

The 3D models used were created using Agisoft Metashape software [50]. The workflow used to reconstruct the 3D model is completely automatic, both in terms of the orientation of the images and the generation and reconstruction of the model. This condition leads to an optimization of processing times ensuring good performance of the machine/software complex. The phases of elaboration were as follows:

1. Align photos (photo alignment): this is the most important phase of the entire photogrammetric process. In this phase, the software aligns our frames with each other, calculating their position in space and reconstructing the so-called gripping geometry. Then, through a process of geometric triangulation, it calculates the position in space of the elements present in the frames. It follows that the quality of this alignment derives from the goodness of the final 3D model. The result of this phase is a sparse point cloud. Following the alignment, the measurement of the control points takes place, to register the block in real-world coordinates, in the reference system defined by the control points.

2. Build Dense Cloud: through this phase, a dense cloud is constructed using dense image-matching algorithms.

3. Extraction mesh: this consists of generating a polygonal model based on the newly created dense cloud. The mesh is a subdivision of a solid into smaller solids of a polyhedral shape.

4. Build texture: this allows us to obtain the 3D representation of the work under investigation.

### 2.4. Automatic Geometrical Information Extraction

From the three-dimensional model obtained, the geometric information (base, beams, deck) was automatically extrapolated, through an appropriate methodology of segmentation of the point cloud. The 3D model obtained was processed to extract relevant geometric information from the model under investigation through a semi-automatic methodology. The procedure extracts the shape from the segmented structural parts and automatically inserts the data into a predefined spreadsheet [51]. Structural parts that are already classified are transformed from a point cloud into a 3D mesh object using Screened Poisson Surface Reconstruction. The workflow of the methodology foresees (following the photogrammetric investigation by drone, and consequent modeling of point clouds and meshes) the automatic transcription of data on a spreadsheet. All components are programmed ad hoc. The basic principle used is the definition of two cutting planes, XY and YZ, to define the section resistant to the structure. The bounding box, as a volumetric element around the object, was created to intersect the cutting plane within the object box. Subsequently, the cutting planes XY and YZ have been configured in a way that the user can define the position of the cutting plane as a percentage of the height, offset distance from the cutting plane and the total number of cutting planes. A preventive check of the flatness and closure of the polyline is carried out. If the polyline is not closed, the algorithm approximates the closure. The developed module was used both on the stacks and on the deck to extract the geometry and insert the data automatically on the spreadsheet.

### 2.5. Implementation on BIM

BIM (Building Information Modeling) and the digitalization of construction processes have been widespread in Italy in recent years. Despite being consolidated, the Italian regulatory framework on BIM does not yet provide specific indications for the infrastructure sector. The succession of the various phases of life of an infrastructural work is a theme of fundamental importance, and it becomes even more important if it is managed using digital and innovative methodologies and tools. The use of the Digital Twin of the work, compatible with VR, AR and MR technologies, represents a virtuous approach in offering navigable and operational virtualizations of infrastructures, for use and consumption by stakeholders. The active involvement of stakeholder engagement in an intervention on the territory is crucial in order to agree on the best design solution for the environment

and for the community. The Digital Twin returns a faithful representation of the possible intervention scenarios and subsequently opens the door to various uses of the model in an integrated manner with the real behavior of the work. Technological evolution is reflected in the vastness of data produced and collected but also, and above all, in the ways in which the data can be transmitted. In recent years, major BIM software houses have been carrying out systems to exploit BIM model information from automatic association between the physical device and its digital twin within dedicated apps. Real-time visual changes on the digital twin within BIM (based on the values of the physical device) allow for precise control even over large environments with tens of thousands of devices. In the specific case of the app developed by the authors, the obtained and segmented photogrammetric models were recomposed within a BIM environment using Revit, which offers the possibility to use nested families [52]. In this way, for example, the 2D sections were created as a mass family and then loaded into an adaptive family model to control the position in relation to other elements. These elements were imported as individual elements in the BIM software (Autodesk Revit) as IFC files, using the entire cloud as a guideline for the placement of the various objects. This process guarantees that the appearance geometrics and material which are closer to the real conformation of each component can be obtained (also ensuring information management) [53].

### 2.6. Historical Images Database and Convolutional Neural Network Processing

The photogrammetric surveys (necessary to acquire the images that are essential both for the 3D model's reconstruction and for the reference database realization to include in the structural degradation evolution analysis) were carried out through the use of drones appropriately programmed for the automated flight phase, meaning they could be repeated over time from the same perspective. The flight plan was programmed on the Drone Harmony app, and the grip points were chosen in such a way as to obtain a GSD < 1 cm/Pixel. The dataset acquired by the drones was processed through a R-CNN method that combined regional proposals with rich features extracted from CNN and was applied to detect deterioration [54]. R-CNN's steps consisted of extracting the region proposal from the input image using a selective search, extracting features using CNN after cropping and detecting the image after the bounding box regression. The detected degradation was calculated using masks [55,56]. The camera captured the image including the homologous points, then the mask was detected using the Region of Interest (ROI) algorithm. Once the acquisitions on the decay are processed and the 3D model of the bridge is built, the structural analysis is carried out. Therefore, the 3D model of the bridge is automatically acquired within the FATA-Next finite element structural software, capable of automatically acquiring all the information on the degradation of the bridge using a CSV file [30]. The CSV file can be subsequently updated each time that the phases described above highlight a change in the degradation conditions detected. Once the datasets were acquired over time (at different times always using the same automated flight plan and, therefore, having defined a priori the waypoints from where acquire the images), it was possible to create, for each area of interest, a sub-dataset useful for visualizing the temporal evolution of the area under examination [56,57], with different purposes, such as degradation evolution, the evolution of installation of services, signage, etc.

### 2.7. Alternative Minim Route Calculation and GPS Range User/Device Interaction

To calculate the alternative minim route, we used the concept of minimum and psychological paths. They do not indicate the achievement of a specific target starting from the optimization of parameters such as time or Euclidean distances (as happens in a classic logistics problem), but they are paths whose routing context appears ergonomic to the human psyche. In other words, we wanted to implement a tool capable of drawing on the virtual map of a GIS application a path that can be performed by the user of the information system without it implying a psychological discomfort to the user. The querying tools present in traditional Geographical Information Systems are unable to cope with those

cases, concerning precisely these new areas in which it is necessary to obtain the existing relationships between cartographic data and environment variables that cannot be sampled in closed form (data on environmental variations, socio-economic variables, thematic areas with blurred contours, etc.). This is mainly due to the impossibility of decomposing all useful queries in terms of a formalism attributable to deterministic–sequential logic and therefore to binary algebra. A type of query intrinsically unsolvable through a Turing Machine concerns the problem of "Minimum Psychological Paths"; given a map and starting from the origin and destination points and considering the user's system, the GIS has to return a path that minimizes the distance of travel (or other logistical parameters) and that appears "comfortable" to the user. The user travels through it, as a function of many parameters that are difficult to define and quantify (such as instinctual reactions to certain situations), related to the context in which the path is connected, or as other properties characterizing the human psyche. In this specific case, a new algorithm has also been implemented for the alternative route calculation (based on the Dijkstra algorithm) that considers the capacity of the infrastructures. The algorithm provides for the recalculation of the route on alternative roads whenever the maximum number of users that can be supported by the infrastructure is reached [58,59]. A function has also been created that, following a markerless tracking process (interaction between GPS coordinates of the device and real coordinates of the study area), among other things, when the coordinates detected by the GPS of the mobile fall within a pre-established range in the construction phase of the app, starts a simulation of the survey or possibility to read a report of the infrastructure [60].

## 3. Results

Referring to the proposed methodology, in order to build an app in VR/AR/MxR for road infrastructure monitoring, consulting information and supporting decisions to managing authorities to determine preventive maintenance interventions, it is necessary to have preliminary:

- Original infrastructure design;
- Any maintenance works;
- Service projects and subservices;
- A 3D survey model (geometric information, state of degradation, etc.);
- A structural model (from which to obtain the different structural responses in the different scenarios).

Specifically, the 3D model was obtained through photogrammetric techniques using UAV (DJI Mavic 2 Pro) with automatic flight at a height of 10 m and centimeter accuracy (GSD < 1), considering the specifications of the DJI Mavic 2 Pro camera. The set of photographic data obtained (1039 photos, 2.5 GB) was processed using the Agisoft Metashape obtaining sparse cloud (Figure 5), dense cloud and a 3D model. The obtained model can thus be used both to execute a virtual inspection and to extract the relevant geometric information about the structure [61–63].

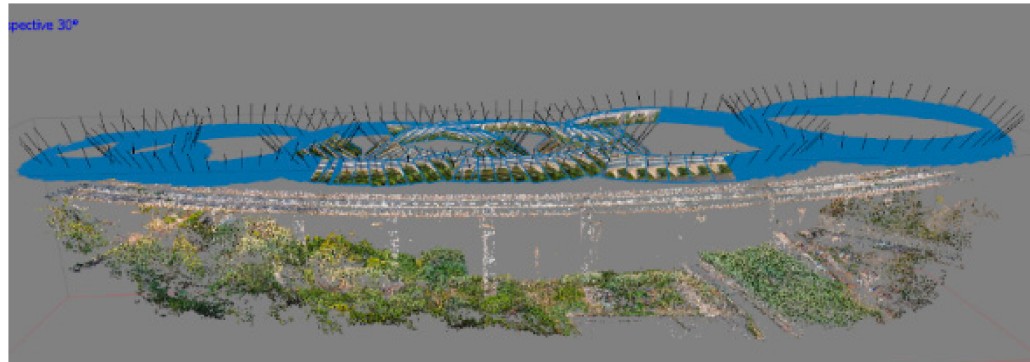

**Figure 5.** Metashape elaboration: scattered cloud from circular photogrammetric survey.

The realization of the structural 3D model can be reconstructed from the project documentation (if available) or through its reconstruction (geometric and degradation from relief, material and components from field inspections) starting from the insertion of the geometric characteristics extracted from the 3D model within the structural model [64]. In the present case, the following was carried out:

- Acquisition of geometry and construction details from the survey and 3D modeling by drone.
- Acquisition of information on the mechanical properties of materials and soils through project documentation.
- Acquisition of "loads" through a system of sensors installed on site.
- Creation of the final structural model using FEM (Figure 6).

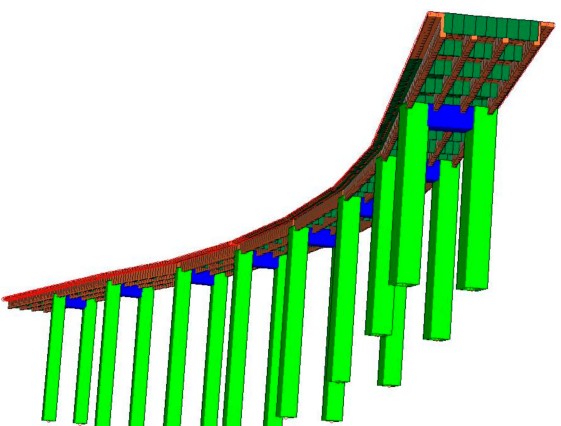

**Figure 6.** FATA-Next elaboration: structural analysis.

The obtained results from the different elaborations (in the scenarios established by the managing authorities) have been automatically classified into different risk classes and, therefore, into intervention priorities; the classes are presented on displays in different colors according to their level of risk, obtained by the analysis and the level of defects highlighted in the various elements [65]. The original plans/designs and additional information were not found in the case study, so they were identified and evaluated following visual inspection. As it is well known in the literature, Augmented and Virtual Reality find different applications in different scientific fields (medicine, mechanics, engineering, etc.). The idea of the authors was to program an app to guarantee infrastructure managers the possibility to transfer the know-how to the various technicians and operators in a simplified, or when necessary, more in-depth way. In fact, the use of the proposed app makes it easy for third parties to inspect and verify what was detected and reported in the data sheets by the technicians, thanks to the potential of Mixed and Augmented Virtual Reality. The app highlights information when the device is near the object of study or when the camera frames it (Figure 7). Therefore, the (above-described) models and information have been integrated into the Unity environment to create an app finalized for road infrastructure monitoring. Thus, the created app for a mobile device allows the user to view the 3D model in AR, visualizing additional information on labels and tags [66], to relate information consultation and to support the managing authorities' decisions. As can be seen from the explanatory flow chart of Figure 6, the application, once started, allows us either to relate the coordinates of the device with those of the artefacts detected within a radius of 10 km connected to the same infrastructure or to display the viaduct model and its information. In the first case, a circumference with a 10 km radius is generated. It identifies a buffering area, which is intersected with the range around the coordinates of other artefacts previously detected and to which specific characteristics have been assigned (structural type, number of spans, height, number of carriageways, functional type). The first selection shows the entire block of identifiable artefacts; then, it is possible to check

only the required characteristics (such as those of the framed bridge). This operation is very similar to the spatial query operations of a GIS. In the second case, the visualization of the built 3D model and the visualization of the information labels start through the "engine" of Unity that superimpose it on the framed object (AR/VR) or reproduce it so as to be viewable directly on the map.

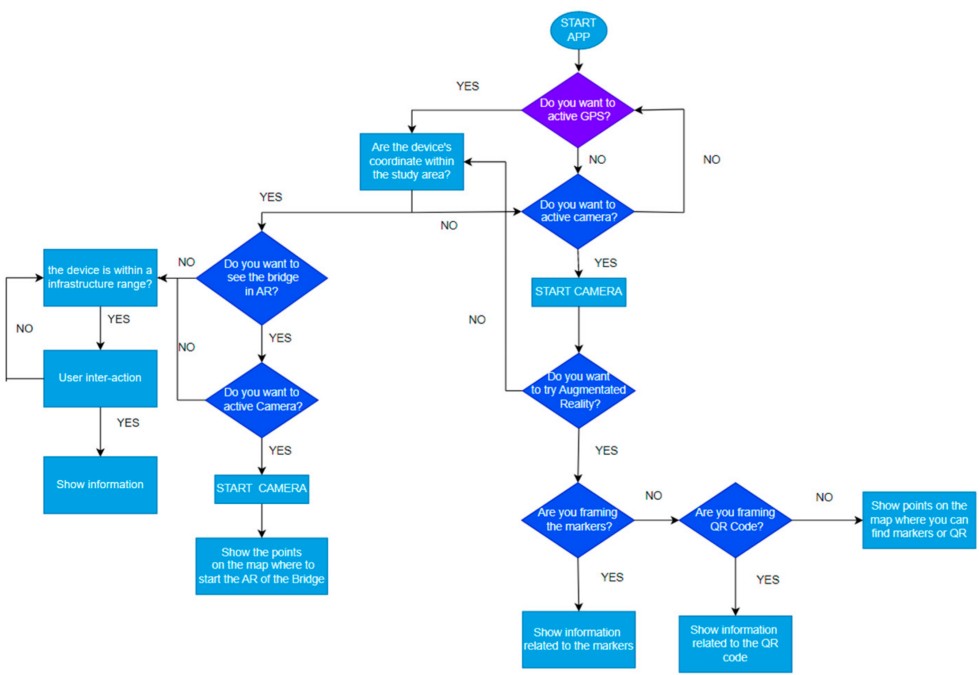

**Figure 7.** App flow chart.

Relative to the first case, the app also allows us to simulate a possible interruption of traffic (total or partial, due to maintenance or accidents) with a redistribution of traffic on other arterial roads, thus suggesting the most convenient graph travel choice for users of the entire road network in which the infrastructure is located, and identifying alternative routes. In particular, we proceeded to assign to each graph (representative of each roadway) a value representative of the maximum vehicular flow capacity and of the average values relating to different daily periods of time (peak hours, night hours, etc.). Whenever the user wants to simulate the intervention on the roadway and, therefore, the reduction (narrowing) or zeroing (closing) of the maximum capacity of the graph, it is verified that the value of vehicular flow (passing in a given hour) exceed this value of maximum "calculated" capacity, and at that point, the traffic is redistributed on the other graphs so as not to exceed the assigned limit value. In the second case, the 3D model of the viaduct is displayed on the map. It is associated with multimedia content, AR/VR or structural analysis identification of degradation or links to other viaducts and bridges on the same road network. For this reason, the use of Unity 3D is aimed at superimposing multimedia content in the form of labels (with which the user can interact) or "objects" (in our case, 3D models and images). In particular, during the implementation of the app, areas defined as "markers" or "particular user actions" are identified, associating to them a specific device action. In particular, in this specific case, the various bridges and viaducts framed from different points of view, the area of the map on which they are located and a well-defined portion of the infrastructure framed were chosen as the main "markers". "User actions" include, among the main ones, the approach of the device within the pre-established range and the framing of particular portions of areas of investigation. All possible user actions, however, have been selected in a drop-down menu in order to facilitate the choice of information by the end user or their remote viewing. There is also the possibility to see the temporal evolution of the 3D model or of a single part of it (zoom in) in Virtual Reality. Thus, the contents can be viewed either through the frame of the infrastructure (and be displayed through labels)

or directly from a drop-down menu [67,68]. In relation to the visualization of content in AR/VR, the flowchart in Figure 6 shows some of the decisions that the user has to make when launching the app. As can be seen from Figure 7, at startup, the user will have to decide whether or not to activate the GPS of the device, so as to be able to define the spatial range within which to identify the additional models made and case studies analyzed. If the user decides to activate the GPS, the app creates a range of 10 km around the user's position and shows any case studies on the map. If the user is within the identified range, then the app offers the possibility to view the 3D model, any user interaction labels/device for displaying information or 3D in VR. In the latter case, the camera is required to be turned on, and the superposition of the 3D model on the framed portion of the bridge is therefore visible [69]. If the user does not activate the GPS, the information in AR and the view of the 3D model in VR will take place directly by camera framing.

In relation to the information collected, in the specific case of monitoring the condition of bridges and viaducts, it should also be noted that both the results of the numerical analysis and the information on the degradation state progress are priority data.

This logical method, therefore, helps to make more information available to the operator than those that are physically in situ, such as the visualization of past events (think of the consequences arising—and recorded—from a decision to interrupt, and therefore from its modulation if the need for a similar intervention arises) [70,71]. Thanks to the marker association, it was possible to directly use a 3D model as a scenario in the VR.

From a functional point of view, the distinctiveness of the implemented app is the capacity to integrate different models, which allows for smooth navigation without blocks or long waiting times for scene loading. In fact, a very important step before inserting the model within the programming environment is the reduction of edges and vertices without modifying the accuracy of the model. A large number of triangles or vertices can reduce performance, especially in devices with performance limitations. If the use of the model is known in advance, it is possible to make choices to reduce triangles, i.e., focusing the reduction on the less important areas with high mesh density. Precise details of the geometric surface and colors of materials can often be replaced by fixing them in normal maps, color maps and ORM (occlusion, roughness and metallic reflection) to save large triangles. It is also important to remove any data that are not necessary for the representation of the model in the displayed scene. Further simplification interventions concern the reduction of the texture, the reduction of the number of graphic instructions per frame (consolidate multiple textures into one) and the simplification of the hierarchies of nodes. Another peculiar aspect of the app is that the location-based AR does not need special markers to identify the location of virtual objects. Location-based Augmented Reality relies on GPS, accelerometer, digital compass and other technologies to identify a device's location and position with high accuracy. The mechanism is quite simple—a mobile app sends queries to sensors. When the app receives information, the application compares it with data about the point of interest (POI) and defines where to place virtual data in the real world. Markerless AR can be divided into outdoor and indoor. Outdoor Augmented Reality uses GPS, while indoor recognizes the current location of a mobile device with beacons.

The application proposed (currently still being defined) allows users to choose the service necessary to implement their exploratory experience. This application is developed to:

- Identify the replacement route in case of closure of the infrastructure. For example, Figure 8 shows an example simulation of the interruption of the road network at the Annunziata viaduct, which can be carried out by the operator. The graph on which the intervention is carried out is colored red, while the main routes on which to sort the traffic are highlighted in green, in the hypothesis of several deviations to reach the maximum capacity set by the infrastructure (Figure 8). Once the capacity of a graph is reached, it is excluded from the calculation of the path.

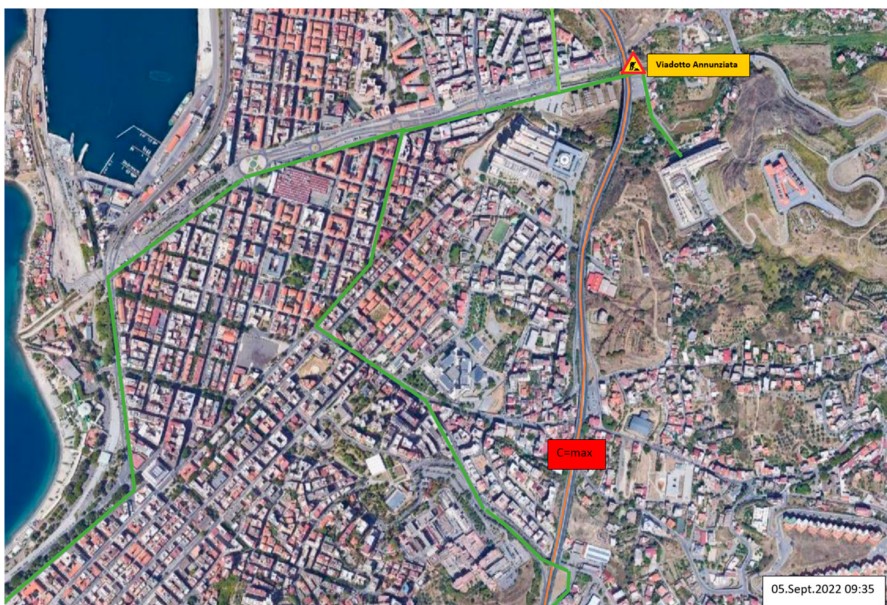

**Figure 8.** AI elaboration for traffic redistribution in the case of closure simulation.

- Offer to the user a virtual tour of the infrastructure and the adjacent area (identifying the boundary conditions and possible interactions with other artefacts on the same network), allowing the user to observe with the possibility to interact with the BIM model through the screen of his device.
  Figure 9a shows the app screenshot where the user can choose the various operations to activate. Figure 9b shows the superimposed 3D model on the viaduct. Figure 9c shows the possibility to explore the photogrammetric 3D model on the map with the ability to rotate or zoom on the model itself (the user can also see the position of other infrastructures in the same area).

- Display the location of services and sub-services, structural elements and the location of degradation. For example, Figure 9 shows the extraction of a single frame where it is reported to a stack that has parts without concrete cover. The archive image acquired during the inspection phases is made available and can be consulted over time. By framing the portions of the viaduct (to which the deterioration images database has been associated), it is possible to visualize whether or not the level of deterioration is subject to aggravation or maintenance (Figure 10a) (the user can also see the deterioration in the 3D maps in Figure 10b).

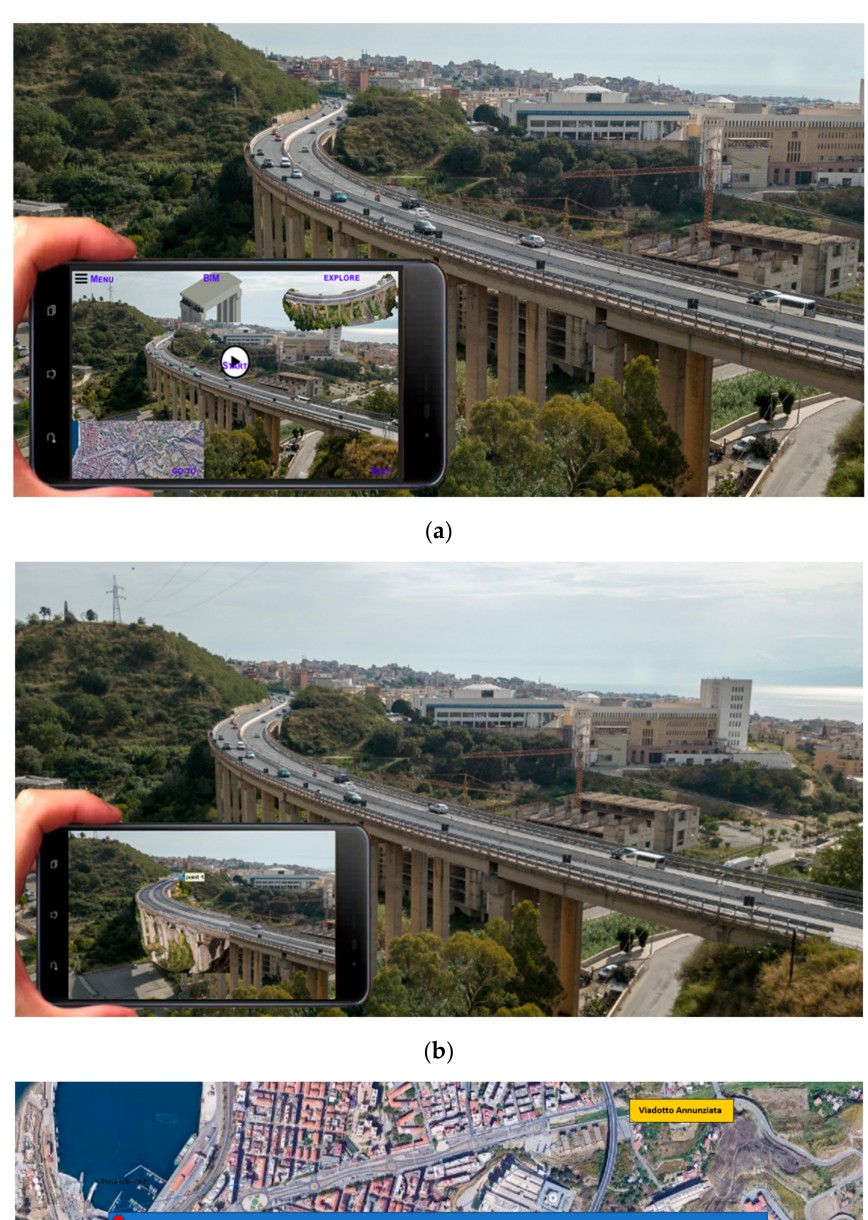

(**a**)

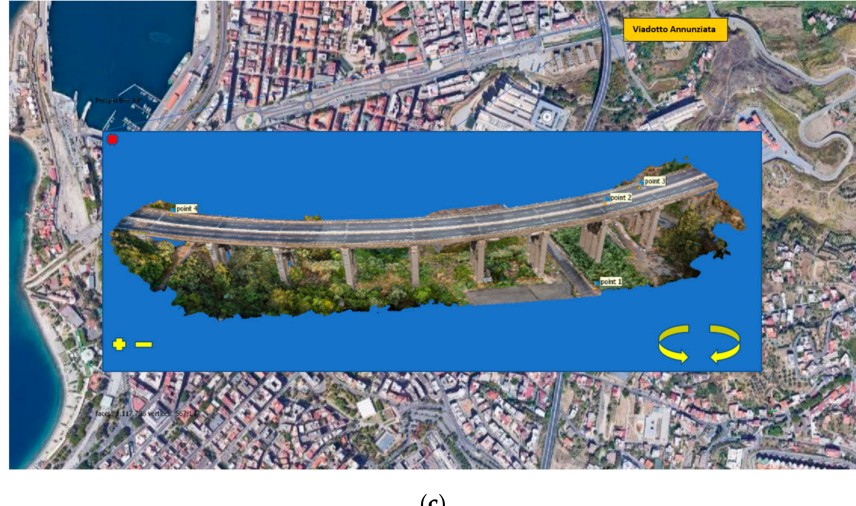

(**b**)

(**c**)

**Figure 9.** (**a**) App home page for viaduct inspection; (**b**) view of the model superimposed on the bridge; (**c**) screenshot of 3D model infrastructure view on the map. The operator can also access it via remote learning on the infrastructure without going to locations.

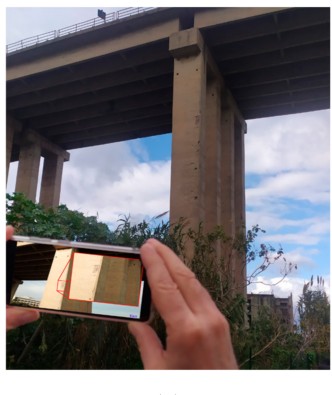
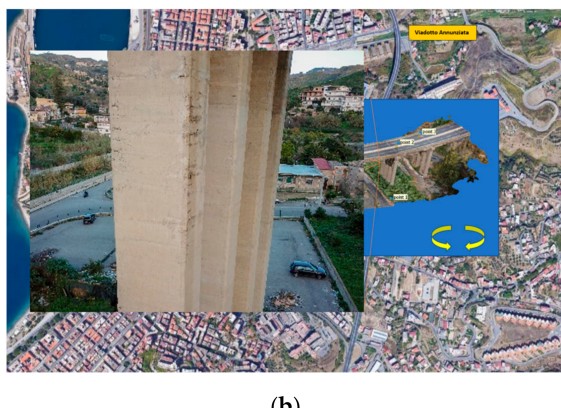

(**a**)  (**b**)

**Figure 10.** (**a**) Inspection's detail of the 3D model in AR for the identification of degradation. (**b**) Visualization of degradation from database image selected from 3D model on the map. The operator can also access it via remote learning on the infrastructure without going to locations.

- Display video and audio associated with the part that is framed by the user. For example, in Figure 11, the results of the structural analyses, which can be viewed by the user, are reported. The device connects to the database through the crossing of two different types of data concerning the verification of the position within a pre-established range (infrastructure position) during the app design phase and the position of the device frame (possibility of multiple infrastructures within the range). The viaduct can be considered a marker, supported by the location-based Augmented Reality.

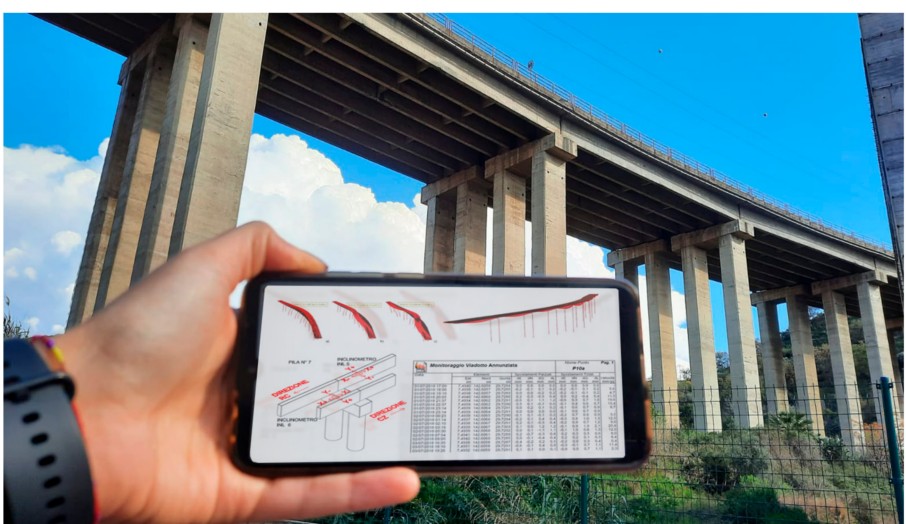

**Figure 11.** Screenshot example of viewable structural results of the framed viaduct under inspection within the coordinate range.

- Highlight the "details of interest" on the framed part and show the tridimensional model of the viaduct; the user can segment and disassemble it into its constituent elements. For example, Figure 12 highlights how the user can view the extraction of the various constituent elements, for the first preliminary analysis necessary for any maintenance interventions.

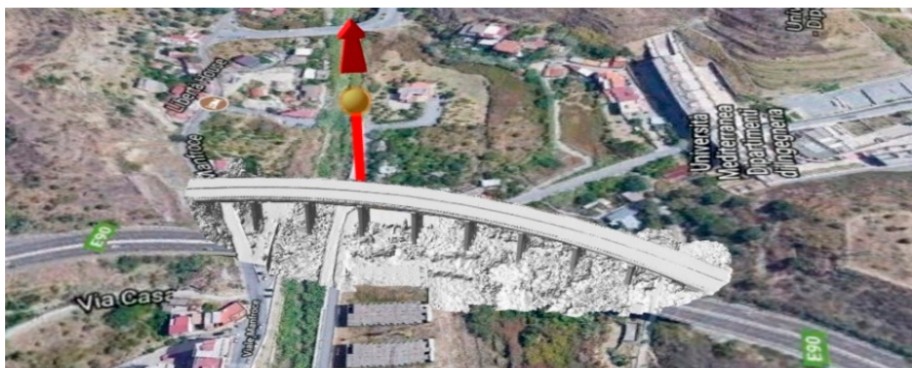

**Figure 12.** Screenshot extraction of the viaduct deck and its individual constituent elements in Virtual Reality.

- Have an immersive experience with Microsoft HoloLens. Figure 13a shows how, in the geomatic laboratory, the user can virtually explore the various phases of a drone's survey. Figure 13b shows a virtual reproduction of the area under investigation. The experience is shared with other participants without headsets through a monitor.

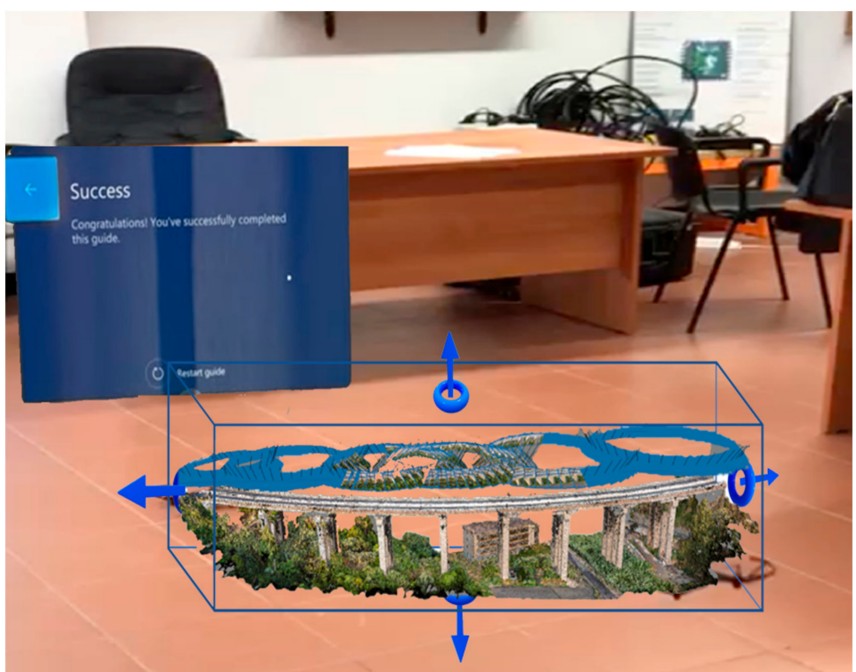

**Figure 13.** Example of Google Lens application to replicate in Mixed Reality and in Virtual Reality the drone's survey.

- Have an immersive experience with Microsoft HoloLens. Figure 14 shows how, on the site, the user can virtually explore the BIM model and explore the single elements that compose it using various phases of the drone's survey. The experience is shared with other participants without headsets through a monitor.

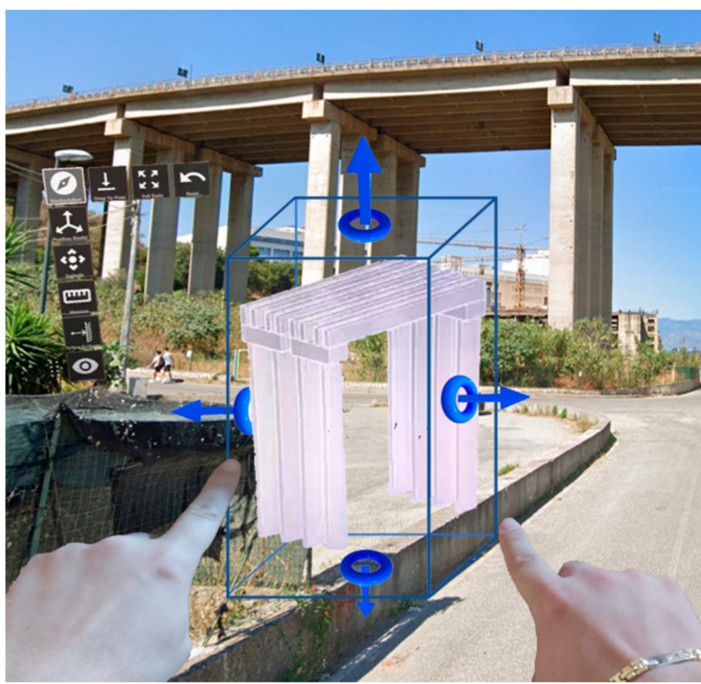

**Figure 14.** Example of Google Lens application to replicate the exploration of a single BIM component on the site.

## 4. Discussions and Future Development

The integration of virtual models (generated using geomatic methodologies) depicting the existing increasingly precise mobile device apps represents the goal to be achieved. In fact, oversimplified models often lose a large amount of information, but models that are too complex need very long loading times, making the virtual experience poor. However, for the applications designed by the authors (purely for the purposes of structural monitoring and/or maintenance of infrastructures), metric precision is very important (consider the comparison of the cracking framework). This would allow us to carry out the processing directly through the app, which would no longer perform the simple role of container and viewer. For this reason, it is necessary to expand studies to improve the user experience, guaranteeing either the weight reduction of the models without losing precision or the speeding up of the buffering phase, regardless of the connection, perhaps dividing the virtual environments into several parts to make playback smoother.

In addition, in the near future, the virtual representation of real environments will represent a business for several companies that are already investing considerable resources in the potential of the Metaverse, which although it is still an ongoing project has some peculiar characteristics that are somewhat delineated.

For this reason, researchers and universities should use resources to improve and regulate this space, through compatible technical standards, protocols, interoperability, digital property, blockchain technology and legislation, with particular reference to the connection between real and digital space through augmented reality and hybrid reality technologies.

## 5. Conclusions

The integration of the classic methods of inspection and analysis of infrastructures with the new technologies of Virtual/Augmented/Mixed Reality is particularly important both for the manager of the road network and/or of the single infrastructure (so as to be able to be supported in maintenance or extraordinary intervention decisions), but also for the individual user who can, on the one hand, benefit from all the information made available by the operator itself and, on the other hand, participate in first person in the collection and transmission of data in real-time. The app, which is still in the definition and

development phase, was carried out in the Unity environment, using models obtained from traditional 3D modeling techniques. However, despite the scaling process that involved the reduction of edges and vertices, this is sometimes too heavy, also considering the non-spread of the 5G network in the Reggio area. The application implemented so far can be easily integrated into a wider context of a smart city, thus generating a single main container in which users also play a primary role and can feel involved in the processes of definition.

**Author Contributions:** Conceptualization, V.B. and A.F.; methodology, V.B. and A.F.; software, V.B. and A.F.; validation, V.B. and A.F.; formal analysis, V.B. and A.F.; investigation, V.B. and A.F.; resources, V.B. and A.F.; data curation, V.B. and A.F.; writing—original draft preparation, V.B. and A.F.; writing—review and editing, V.B. and A.F.; visualization, V.B. and A.F.; supervision, V.B. and A.F.; project administration, V.B. and A.F.; funding acquisition, V.B. and A.F. All authors have read and agreed to the published version of the manuscript.

**Funding:** This research received no external funding.

**Data Availability Statement:** Data sharing is not applicable to this article.

**Conflicts of Interest:** The authors declare no conflict of interest.

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
