# Peer review of "Viaduct and Bridge Structural Analysis and Inspection through an App for Immersive Remote Learning"

_electronics, doi:10.3390/electronics12051220_

Round 1

Reviewer 1 Report

This study presented an integration of immersive 3D technologies, geomatics, survey, and structural monitoring techniques for better visualization and understanding of the viaduct and bridge structure. The study emphasized immersive remote learning, and it is not augmented or mixed reality. The visualization technology adopted in this study is only Virtual Reality. The authors should investigate the visualization technologies in detail since you are going to talk about them. The difference should be demonstrated and not confused with the concepts of Virtual Reality, Augmented Reality, and Mixed Reality.

1.       The title is suggested to get rid of the augmented reality description.

2.       Only in the introduction part, augmented reality and mixed reality can be retained. Reconstruct the manuscript and get rid of the wrong description of augmented reality and mixed reality.

3.       There is no sub-graph in Fig. 7 while using Fig. 7 a, b in the description (line 79).

4.       The definition of “Virtual Reality” in line 123 is wrong.

5.       The manuscript should describe the 3D modeling and interaction processes in detail.

Author Response

Dear reviewer,
thank you very much for your suggestion, we tried to modify according with your observation some parts of the paper. We made some integration in some parts and rewrote others. In Particular, in response to your observations, we report our comment point by point.

  1. The title is suggested to get rid of the augmented reality description.
    A further and more in-depth description of augmented reality has been included in the paper
  2. Only in the introduction part, augmented reality and mixed reality can be retained. Reconstruct the manuscript and get rid of the wrong description of augmented reality and mixed reality.
    The different descriptions have been reworded
  3. There is no sub-graph in Fig. 7 while using Fig. 7 a, b in the description (line 79).
    It's a refuse has been eliminated
  4. The definition of “Virtual Reality” in line 123 is wrong.
    The different descriptions have been reworded
  5. The manuscript should describe the 3D modeling and interaction processes in detail.
    The description has been inserted as suggested by reviewer

Reviewer 2 Report

The article tackles an interesting topic. Immersive technologies, such as virtual reality (VR), augmented reality (AR), and mixed reality (MR), are digital technologies that can be used to enhance or augment the user's perception of the real world. These technologies can potentially be used in a variety of fields, including engineering, architecture, and construction, to improve visualization and understanding of complex structures and systems.

The authors are exploring the use of immersive technologies to assist with the maintenance of infrastructure, specifically viaducts and bridges. They have developed an application in Unity 3D that allows users to explore the structure in a virtual environment and access all relevant information related to the structure through 3D analysis. This can potentially be a more efficient and safe way to perform inspections or analyses of the structure, as it allows professionals to remotely access and visualize the structure without the need to physically be present.

In addition to improving the visualization and understanding of maintenance interventions, the authors also argue that the use of immersive technologies can help managing institutions make more informed decisions about infrastructure maintenance. By providing a more comprehensive view of the structure and its condition, professionals can potentially identify problems or issues more quickly and accurately, which can help reduce post-intervention costs. Overall, the authors suggest that the integration of immersive technologies with other techniques, such as geomatics and structural monitoring, can provide a valuable tool for the maintenance and management of infrastructure.

The possibilities related to the idea for the application look very good. However, the article focuses too much on generalities. Too little has been presented about the specific achievements of the authors in this topic.

In its current form, the article is more an idea for research than a presentation of research results.

That is why I strongly believe that the article should be vastly improved before publishing.

To provide more evidence and support for their claims, the authors could consider including more detailed descriptions and examples of how their developed app was used in the analysis and inspection of the viaduct in Reggio Calabria. This could include specific examples of how the app was used to visualize and access information about the structure, as well as any observations or insights that were gained through the use of the app.

In addition, the authors could consider including actual footage from the app, either through the use of screenshots or video demonstrations. This would allow readers to see firsthand how the app works and how it can be used to visualize and understand the structure. By providing more concrete examples and evidence, the authors can better demonstrate the effectiveness and utility of their app in assisting with the maintenance and management of infrastructure.

Additional editorial comments:

I suggest that the authors explain the abbreviations used in the article each time they appear for the first time. For example, “ANAS” in the first sentence of the article is explained only at the beginning of page 3.

The language of the article should be checked again.

Spelling mistakes, e.g., “reserch’s” instead of research, “whit” instead of with.

Syntax: Sentences like: “The Managing Institution in charge of the maintenance of the road infrastructure (Municipal, ANAS etc.), in the event that the structure is recent they have available the executive projects and therefore can make investigations by comparing with the project data [1];”

or

“Unity platform programming is based on “Game Objects” [25, 26] whit or without graphic representation. To them were associated “Mono-Behavior” scripts, that allow verifying the presence of new data in the Buffer.”

need to be rewritten. Their syntax is strange and sounds like a literal translation from another language. I suggest dividing them into shorter sentences which are clearer.

Additionally, the English language in the article should be consistent. i.e., once the spelling follows the rules of the US English, and once UK English. E.g., visualize vs. visualisation.

Author Response

Dear reviewers,
thank you very much for your suggestion, we tried to modify according with your observation some parts of the paper. We made some integration in some parts and rewrote others. In Particular in response to your observations, we report our comment point by point.

The possibilities related to the idea for the application look very good. However, the article focuses too much on generalities. Too little has been presented about the specific achievements of the authors in this topic.
Referring also to the suggestions of the other reviewers, we proceeded with the insertion and a more detailed description of the methodologies with the emphasis of the specific results obtained and obtainable

In its current form, the article is more an idea for research than a presentation of research results.
The results are reported with individual screenshoots, however we proceeded to carry out a further integration as suggested by the reviewers.

That is why I strongly believe that the article should be vastly improved before publishing.
The article was vastly improved before publishing as suggested by other reviewers.

To provide more evidence and support for their claims, the authors could consider including more detailed descriptions and examples of how their developed app was used in the analysis and inspection of the viaduct in Reggio Calabria. This could include specific examples of how the app was used to visualize and access information about the structure, as well as any observations or insights that were gained through the use of the app.
Referring also to the suggestions of the other reviewers, we proceeded with the insertion of a more detailed description, adding additional images and specifying the potential of the app in the materials and methods section.

In addition, the authors could consider including actual footage from the app, either through the use of screenshots or video demonstrations. This would allow readers to see firsthand how the app works and how it can be used to visualize and understand the structure. By providing more concrete examples and evidence, the authors can better demonstrate the effectiveness and utility of their app in assisting with the maintenance and management of infrastructure.
in addition to the 7 screen shoots already present in the text, We have added two more screens as suggested

 Additional editorial comments:

I suggest that the authors explain the abbreviations used in the article each time they appear for the first time. For example, “ANAS” in the first sentence of the article is explained only at the beginning of page 3.

"Azienda Nazionale Autonoma delle Strade" (A.N.A.S.)  is an Italian joint-stock company that deals with road infrastructure and manages the network of state roads and highways of national interest.

The language of the article should be checked again.
We checked again the language

Spelling mistakes, e.g., “reserch’s” instead of research, “whit” instead of with.
We checked again the language

Syntax: Sentences like: “The Managing Institution in charge of the maintenance of the road infrastructure (Municipal, ANAS etc.), in the event that the structure is recent they have available the executive projects and therefore can make investigations by comparing with the project data [1];”
We rewrote it

or

“Unity platform programming is based on “Game Objects” [25, 26] whit or without graphic representation. To them were associated “Mono-Behavior” scripts, that allow verifying the presence of new data in the Buffer.”
We rewrote it

need to be rewritten. Their syntax is strange and sounds like a literal translation from another language. I suggest dividing them into shorter sentences which are clearer.
We rewrote them

 Additionally, the English language in the article should be consistent. i.e., once the spelling follows the rules of the US English, and once UK English. E.g., visualize vs. visualisation.
We corrected it

Reviewer 3 Report

The article is more a case of application development than research. It suffers from a number of issues that make it recommend rejection for publication: it presents serious evidence of structure, in the introduction there is no evidence of the knowledge gap to be covered by the research, and there is no mention of part of the methodology. The section on materials and methods is totally general and does not explain the methodology. In the results, the methodology is explained in a very superficial way. The explanation of the application is reduced to a scheme of information flow, without indicating how the connection of the anomaly information, numerical analysis and virtual reality is carried out. Finally, in the results section, a series of statements are made in the figures that are totally different from what appears in them.

Further details of the above are given below:

The introduction describes part of the methodology.

The introduction does not provide evidence of the existing knowledge gap, which shows that the research proposal is necessary. There is a significant amount of research on bridge assessment and inspection, bridges and BIM, virtual reality and bridges, etc., and a minimum number of references (ten) have been used in the introduction, some of them on very collateral topics such as BIM and tourism, BIM and heritage buildings.

The title of the article refers to the application of augmented reality for immersive learning, but the introduction does not say anything about learning.

The introduction is limited to setting out the methodology of the work, the fundamentals of BIM technology, which are widely known, with a poor use of references. For example, reference [2] is focused on virtual reality, not on sensors as indicated in the text. In the section where reference [10] is used, they are talking about the 4D of BIM, however the referenced research does not make any mention of BIM in the whole text.

References 12, 13, 14, 15, 16, 17, 18 and 19 appear for the first time in the results.

The proposed methodology, therefore, concerns the design and subsequent implementation of a system to monitor infrastructures".

Part of section 2 is not methodology, it is general comments, with statements that require justification and how it can be carried out in this research. If it is not carried out it should not be part of the methodology.

For example, paragraph 2 states "In the case of existing works, structural monitoring integrates the data used in the design phase. It is also possible to determine the response of the structure combined with data obtained from surveys in situ and data from tests carried out in the laboratory. In this case, therefore, we are aware of the response and loading actions, while what we want to determine is the pattern. It could be said that in the first case we deal with a direct problem, while in the second case we deal with the opposite problem".

In paragraph 2 it says "To validate the final model obtained, the stresses detected are then compared with those found during the in-situ tests". How is the model compared with the in-situ tests? What kind of tests allow such a comparison? On the other hand, current methods of evaluating existing structures are based on limit states where stresses not strains are evaluated.

The comparison between the model (at the present time and the simulated one at the time n) allows to identify any anomalies; therefore, it's possible to decide a priori how to intervene to prevent the occurrence of any damage, carrying out preventive maintenance activities that clearly reduces and spreads the funds available on the entire road infrastructure". This statement requires further justification: any anomaly, how is it possible to predict a priori how to prevent damage: what damage, with what reliability, with what data to feed this model?

In rows 123-124 it says "As Known a Virtual reality system is generated by the combination of the real scene (seen by the user) and the pre-registered digital virtual scene, which can be significantly influenced by the behaviour of the active subject". As the name indicates, in virtual reality the scene is virtual, digital, designed.

In addition to all of the above, not a single line is devoted to explaining the specific materials and methods of this research, nor the process for creating the scenarios, nor how the finite element analysis has been integrated into virtual or augmented reality, etc., etc.

Results. Figure 8 states "Viaduct inspection frame in Augmented Reality. Operator can access remote learning on infrastructure without going to locations. "and what appears is a superimposition of a computer image with a 3D image of the model. For augmented reality to exist, it is necessary to superimpose the digital model on reality, so the operator will have to be in situ, not remotely, which is what could be done with virtual reality.

Figure 9 . Inspection's detail of the 3D model in AR for the identification of degradation. This is not augmented reality, it is again an overlay of digital images, and a photograph of a part of a bridge pier.

Figure 10. Example of structural results viewable in augmented reality. In this case, an image of reality does appear, but what appears in the application on the tablet are several models with analytical results and a table. The same can be seen on any computer screen without the need to go to the bridge.

Figure 11. Extraction of the viaduct deck and its individual constituent elements in 305 virtual reality. This is not virtual reality, it is the superimposition of a googlemaps or similar image and a 3D model that appears to be extracted from the point cloud mesh.

Figure 12 has the same text as Figure 11.

Author Response

Dear reviewer,
thank you very much for your suggestion, we tried to modify according with your observation some parts of the paper. We made some integration in some parts and rewrote others. In Particular, in response to your observations, we report our comment point by point.

The article is more a case of application development than research.

We agree that the article mainly concerns an applicative activity deriving from the combination and experimentation of different methodologies applied to completely different contexts than the one proposed. The aim of the research is in fact to bring out the potential of the integration of these methodologies aimed at monitoring bridges and viaducts, in order to support the technicians of the managing bodies as a container of information, made available at any time (even during on-site inspections) as better highlighted in the materials and methods section.

It suffers from a number of issues that make it recommend rejection for publication:

  • it presents serious evidence of structure, in the introduction, there is no evidence of the knowledge gap to be covered by the research, and there is no mention of part of the methodology.

We provided to integrate the note with what is required

  • The section on materials and methods is totally general and does not explain the methodology.
    In the results, the methodology is explained in a very superficial way.
    We have integrated with the explanation of the methodology

  • The explanation of the application is reduced to a scheme of information flow, without indicating how the connection of the anomaly information, numerical analysis and virtual reality is carried out.
    We have integrated with thendication how the connection of the anomaly information, numerical analysis and virtual reality is carried out.
    .
  1. Finally, in the results section, a series of statements are made in the figures that are totally different from what appears in them.
    We have integrated the images with additional screenshoots.

Further details of the above are given below:

  • The introduction describes part of the methodology.
    We have integrated with what is required

  1. The introduction does not provide evidence of the existing knowledge gap, which shows that the research proposal is necessary.
    We have integrated with what is required
    There is a significant amount of research on bridge assessment and inspection, bridges and BIM, virtual reality and bridges, etc., and a minimum number of references (ten) have been used in the introduction, some of them on very collateral topics such as BIM and tourism, BIM and heritage buildings.
    The methodologies used have been detailed in the body of the text, however, it is emphasized that the HBIM and Bim and Turism procedure as well as the others have been used for the realization of the present work as also indicated in the additions of the paragraph materials and methods

The title of the article refers to the application of augmented reality for immersive learning, but the introduction does not say anything about learning.
The goal of the methodology and therefore of the application created is precisely to want to provide technicians and end users with a tool that can serve as a source of knowledge (and therefore a single collector of information, updatable) through the fundamental characteristics of immersive reality. In any case, what is required has been added in the introduction.

The introduction is limited to setting out the methodology of the work, the fundamentals of BIM technology, which are widely known, with a poor use of references.
We have integrated with what is required

For example, reference [2] is focused on virtual reality, not on sensors as indicated in the text. In the section where reference [10] is used, they are talking about the 4D of BIM, however the referenced research does not make any mention of BIM in the whole text.

We have corrected and re-examined the error that has been reported.

References 12, 13, 14, 15, 16, 17, 18 and 19 appear for the first time in the results.
We have proceeded to introduce them in line with what has been observed

The proposed methodology, therefore, concerns the design and subsequent implementation of a system to monitor infrastructures".

Part of section 2 is not methodology, it is general comments, with statements that require justification and how it can be carried out in this research. If it is not carried out it should not be part of the methodology. For example, paragraph 2 states "In the case of existing works, structural monitoring integrates the data used in the design phase. It is also possible to determine the response of the structure combined with data obtained from surveys in situ and data from tests carried out in the laboratory. In this case, therefore, we are aware of the response and loading actions, while what we want to determine is the pattern. It could be said that in the first case we deal with a direct problem, while in the second case we deal with the opposite problem".

The proposed methodology has been designed for its application on an entire territory managed by several managing bodies in such a way that an order of priority of interventions can be established in such a way as to avoid overlaps and waste. In this regard, it is not said that on different bridges it is necessary to carry out all the phases. In the present case, for example, no documents were found and therefore the whole investigation developed.

In paragraph 2 it says "To validate the final model obtained, the stresses detected are then compared with those found during the in-situ tests". How is the model compared with the in-situ tests? What kind of tests allow such a comparison? On the other hand, current methods of evaluating existing structures are based on limit states where stresses not strains are evaluated.

Several checks are carried out. For example, the actions acting on the model, from the project were defined through the TGM (average daily traffic from bibliography) while in situ the actual number of vehicles (pattern recognition system in the various frames) are actually counted (during peak hours therefore of maximum load) or the reduction of the section measured through photogrammetric techniques and validated through in situ measurements,  clearly it can lead to a change in structural behaviour (degradation). Therefore, with regard to the response of the structural model, the various parameters are used to update the models already built and obtain answers more congruous with the state of affairs. (activity that nowadays is still carried out in a non-automated way but through the use of operators who visually inspect bridges and viaducts with destructive investigations).

The comparison between the model (at the present time and the simulated one at the time n) allows to identify any anomalies; therefore, it's possible to decide a priori how to intervene to prevent the occurrence of any damage, carrying out preventive maintenance activities that clearly reduces and spreads the funds available on the entire road infrastructure". This statement requires further justification: any anomaly, how is it possible to predict a priori how to prevent damage: what damage, with what reliability, with what data to feed this model? Considering that the deterioration of the infrastructure varies over time with a non-linear trend, understanding the extent of degradation and its evolution allows you to create different models (predictive, both from the point of view) and therefore intervene to restore the materials or in any case reduce the damaged parts allows to avoid the increase in costs.

In rows 123-124 it says "As Known a Virtual reality system is generated by the combination of the real scene (seen by the user) and the pre-registered digital virtual scene, which can be significantly influenced by the behaviour of the active subject". As the name indicates, in virtual reality the scene is virtual, digital, designed.
We rewrote the sentence.

In addition to all of the above, not a single line is devoted to explaining the specific materials and methods of this research, nor the process for creating the scenarios, nor how the finite element analysis has been integrated into virtual or augmented reality, etc., etc.
we rewrote this part

Results. Figure 8 states "Viaduct inspection frame in Augmented Reality. Operator can access remote learning on infrastructure without going to locations. "and what appears is a superimposition of a computer image with a 3D image of the model. For augmented reality to exist, it is necessary to superimpose the digital model on reality, so the operator will have to be in situ, not remotely, which is what could be done with virtual reality.

Fig. 8 show the exploration of the photogrammetric model with the ability to rotate or zoom on the model itself. Clearly, the same operation can be done by framing on the bridge in situ. We add a new image that show its.

Figure 9. Inspection's detail of the 3D model in AR for the identification of degradation. This is not augmented reality, it is again an overlay of digital images, and a photograph of a part of a bridge pier.

We agree. We added a different screen, also framing the bridge. The screen that is actually used on the paper, shows the possibility to see a part of the degradation from the 3D model.

Figure 10. Example of structural results viewable in augmented reality. In this case, an image of reality does appear, but what appears in the application on the tablet are several models with analytical results and a table. The same can be seen on any computer screen without the need to go to the bridge.

We agree that the image can be seen on any computer screen without the need to go to the bridge, but the application shows how the user, once arrive close to the bridge under inspection and frame on it, he can click on different label that appears on it and see on the screen the desired information. We add a previous screenshot of the app.

Figure 11. Extraction of the viaduct deck and its individual constituent elements in 305 virtual reality. This is not virtual reality, it is the superimposition of a googlemaps or similar image and a 3D model that appears to be extracted from the point cloud mesh.

It is not a superimposition of Google maps or similar image and a 3D model. (For sure, the model is extracted from the point cloud model). The figure shows the possibility to see a 3D model or its detail of interest once the bridge is chosen. The user can see the model, o zooms in on it. It is a virtual reality because once is the user frame the bridge, the app shows the position on the maps and its model. We would remember that obviously there is also the possibility to select the bridge from the database app, without framing it (the application is finalised to have a “big database” for each infrastructure that can be used for different applications).

Figure 12 has the same text as Figure 11.

We rewrote the text of figure 12

Reviewer 4 Report

The article shows an interesting topic in which the integration of immersive technologies (VR, AR and MR) with survey and structural monitoring contributes to infrastructure management and maintenance. The case study is a viaduct located in Reggio Calabria. The topic could be particularly interesting in several areas, not only infrastructural but also architectural for ancient or locally present buildings.

However, I would recommend a more in-depth study of the state of the art by analysing what methodologies are proposed today and what is the innovation or a particularly in-depth analysis brought with this study.

I would also suggest a more detailed analysis of some parts related to the methodology developed, the instrumentation and the software used, as well as an in-depth look at how the integration was implemented within Unity.

It is also necessary to review the text as it has some formatting errors, such as in Keywords; at line 96 absence of a point; space at line 257, probably the reference to Figure 1 is incorrect at line 263. I suggest revising the wording of the text without starting a new paragraph so many times, as well as revise some images that are presented with low resolution (such as Figure 1). I also suggest standardising the papers in the references following the editorial guidelines.

In conclusion, once the above additions are made, the article could become an interesting scientific paper on data integration and infrastructure management. I look forward to reading the completed article.

Author Response

Dear reviewers,
thank you very much for your suggestion, we tried to modify according with your observation some parts of the paper. We made some integration in some parts and rewrote others. In Particular, in response to your observations, we report our comment point by point.

However, I would recommend a more in-depth study of the state of the art by analysing what methodologies are proposed today and what is the innovation or a particularly in-depth analysis brought with this study.
We improve the state of art.

I would also suggest a more detailed analysis of some parts related to the methodology developed, the instrumentation and the software used, as well as an in-depth look at how the integration was implemented within Unity.
We rewrite it according with your suggestion.

It is also necessary to review the text as it has some formatting errors, such as in Keywords; at line 96 absence of a point; space at line 257, probably the reference to Figure 1 is incorrect at line 263. I suggest revising the wording of the text without starting a new paragraph so many times, as well as revise some images that are presented with low resolution (such as Figure 1). I also suggest standardising the papers in the references following the editorial guidelines.
We replaced the fig.1.

We corrected the formatting errors according to the editorial guidelines.

Round 2

Reviewer 1 Report

After the revision, the paper is qualified to be published.

Author Response

Thank you very much for your contribution, we provided to check English language and style 

Reviewer 2 Report

I believe my comments from a previous review have not been implemented satisfactorily.
I stand by my previous review.
The submitted article is a rather interesting idea for an article, not the work ready to be published.
It still lacks evidence of the existence of the described application. The authors present only a series of edited photos that look like an idea for an application, not a ready-made solution.
Evidence of the results achieved must be more conclusive.

Author Response

  • I believe my comments from a previous review have not been implemented satisfactorily. I stand by my previous review.

Although we have tried in the first revision to process your requests, we have nevertheless added further additions, both in relation to the best description of the methodology, and in terms of presentation of application results (to provide more evidence and support for the methodology), trying to better highlight the potential of the app, both by inserting additional screens to allow readers to see firsthand how the app works and how it can be used to visualize and understand the structure.

  • The submitted article is a rather interesting idea for an article, not the work ready to be published.

Further additions have been added to meet the requests; we tried to avoid to go into too much detail,   that, in our opinion would, not lead to any increase, since they are now known and present in the literature to which reference has been made. In fact,  (nowdays it is almost a game and accessible to everyone even the implementation of any code that can be done by functions and buttons  already present in commercial dedicated software), from a point of view of the app implementation it is quite clear that the main difficulty in creating an app in AR / VR lies precisely in its conception, in the choice of data to be used and in the refinement of the results (in addition of course to the optimization of the spaces necessary for data storage). That said, following the comments, we have integrated the original note by better detailing them where required.

  • It still lacks evidence of the existence of the described application. The authors present only a series of edited photos that look like an idea for an application, not a ready-made solution. Evidence of the results achieved must be more conclusive.

In order to respond to the observation received, further paragraphs have been added to highlight the procedure followed for the app realization; moreover, we observe that the screenshots are deliberately reported to highlight both the importance of the idea developed in the app (trying to understand the application implications of the methodology) and both why the outdoor lighting conditions do not allow to frame the real scenario and the phone with adequate resolutions to be presented optimally in a paper. Aas also requested by another reviewer, additional images have been inserted to better highlight the use of the device with augmented reality, making the evidence of the results obtained more conclusive.

Reviewer 3 Report

After reviewing the article, it shows methodological shortcomings that do not describe the process of information flow generation, as well as conceptual shortcomings.

Some issues:

In figure 8 and 9 there is an overlapping of images that do not demonstrate the application of augmented reality: seeing in the same image the reality and the digital model.  This was already pointed out in the previous review, to which the authors responded:
We agree. We added a different screen, also framing the bridge. The screen that is actually used on the paper, shows the possibility to see a part of the degradation from the 3D model.
However, the image is kept and another one is added, which is not augmented reality either.

Figure 10 does not add anything to the augmented and virtual reality research, it is a view of the bridge in the background with a tablet on which structural analysis images.

Figure 11 is not virtual reality, it is superimposed on an image from Google maps or similar with a 3D model obtained from a point cloud. Virtual reality would have to take the bridge model from BIM with information.

Author Response

  • After reviewing the article, it shows methodological shortcomings that do not describe the process of information flow generation, as well as conceptual shortcomings.

Although we believe that the process of information flow generation was already present in the first version of the article, however, following the comments received we have taken steps to strengthen and integrate it, and we have added further information avoiding however to report notions and details that are now known in the literature.
With regard to the lack of conceptual shortcomings, although it is difficult for us to further deepen the issues addressed, especially in this case, further information has been included to better explain how the authors have come to the integration of the different methodologies and therefore to the realization of the Virtual Mixed Augmented Reality app presented in the note.

Here are some examples of other apps in AR and VR that individually report the methodologies described, developed, integrated and implemented in a single app by the authors.

- PeakLens - Mountain Identification Android Mobile App

- PeakVisor AR App and Site for Mountains Explorers

- Monte Castiglione - PeakFinder

- The global Summit Log for the iPhone and Android Smartphones (peakhunter.com)

- Mountain Live Explorer - ALPI - Android - App italiane

- Pl@ntNet identify (plantnet.org)

- Una Community per Naturalisti · iNaturalist

- Pokémon GO | Videogiochi e app (pokemon.com)

  • In figure 8 and 9 there is an overlapping of images that do not demonstrate the application of augmented reality: seeing in the same image the reality and the digital model.  This was already pointed out in the previous review, to which the authors responded: "We agree. We added a different screen, also framing the bridge. The screen that is actually used on the paper, shows the possibility to see a part of the degradation from the 3D model."
    However, the image is kept and another one is added, which is not augmented reality either. Figure 10 does not add anything to the augmented and virtual reality research, it is a view of the bridge in the background with a tablet on which structural analysis images. Figure 11 is not virtual reality, it is superimposed on an image from Google maps or similar with a 3D model obtained from a point cloud. Virtual reality would have to take the bridge model from BIM with information.

In order to respond to the comments received, observing that the screenshots referring to the figures indicated had been deliberately reported both to highlight the importance of the idea developed in the app (trying to understand the application implications of the methodology), both because the outdoor lighting conditions do not allow to frame the real scenario and the phone with adequate resolutions to be presented optimally in an article, this premised, as requested also by another reviewer, additional images (8 e 9) have been inserted that better highlight the use of the device with augmented reality. Regarding Fig 10 and fig 11, we should prefer don't remove them because we are sure that this part of the application works using AR, and for this reason, we tried to better explicate the process to generate that part of app, and why it is AR. Anyway if you think that we have to remove them just let us know.

Reviewer 4 Report

The additions are significant and enrich the content of the paper. However, I suggest improving the organisation and drafting of the text. In the Introduction, reference could be made to some similar integrated platforms already created to highlight what has been developed so far, the state of art only refers to classical methods of monitoring the bridge and with sensors application. In the Materials and methods section, for ease of reading, there could be several sub-chapters in which each aspect could be analysed: the stages of the drone survey, and the point cloud.

I would avoid starting a new paragraph many times and leaving blank lines. I suggest revising the GSD values of the photogrammetry (once 1.5 cm and the other 1 without units). I also recommend paying attention to the position of captions, which should be avoided on pages other than that of the image, as in the case of Fig. 8. Pay attention to superfluous or incomplete sentences 199 and lines 48 - 296 where a new paragraph begins. Finally, I once again invite you to review the References, which must be written as stated in the Electronics mdpi journal rules; comma points, italicised characters, pages... are missing.

Author Response

Dear reviewer, thank you for your useful observation we follow them and made corrections to the text.

  • However, I suggest improving the organisation and drafting of the text.

We have restructured the organization according to further indications

  • In the Introduction, reference could be made to some similar integrated platforms already created to highlight what has been developed so far, the state of art only refers to classical methods of monitoring the bridge and with sensors application.

We have included in the text the following: "the results of the traditional methodologies used today are not adequately grouped and can be consulted on integrated platforms, There are clearly web platforms and GIS systems but their use is aimed at representing results related to a single problem."

  • In the Materials and methods section, for ease of reading, there could be several sub-chapters in which each aspect could be analysed: the stages of the drone survey, and the point cloud.

We have reinserted the paragraphs according to indications.

  • I would avoid starting a new paragraph many times and leaving blank lines.

We avoid starting a new paragraph many times as request

  • I suggest revising the GSD values of the photogrammetry (once 1.5 cm and the other 1 without units).

We check and correct them. There was a refuse.

  • I also recommend paying attention to the position of captions, which should be avoided on pages other than that of the image, as in the case of Fig. 8.

We check and correct them

  • Pay attention to superfluous or incomplete sentences 199 and lines 48 - 296 where a new paragraph begins.

We check and correct them

  • Finally, I once again invite you to review the References, which must be written as stated in the Electronics mdpi journal rules; comma points, italicised characters, pages... are missing.

We check and correct them

Round 3

Reviewer 2 Report

Thank you for improving the article.

Author Response

Thank you for your contribution.

Reviewer 3 Report

The answers provided still do not show the progress that the research proposed does, and therefore this reviewer is sorry to say that the document is not of sufficient quality to be published.

Author Response

Dear reviewer,
we made some other changes. 

We hope that yuo can find the final verison improved.